# A multimodal sensing ring for quantification of scratch intensity

Akhil Padmanabha [1✉], Sonal Choudhary[2], Carmel Majidi [1,3,4] & Zackory Erickson [1,4]

## Abstract

**Background** An objective measurement of chronic itch is necessary for improvements in patient care for numerous medical conditions. While wearables have shown promise for scratch detection, they are currently unable to estimate scratch intensity, preventing a comprehensive understanding of the effect of itch on an individual.

**Methods** In this work, we present a framework for the estimation of scratch intensity in addition to the detection of scratch. This is accomplished with a multimodal ring device, consisting of an accelerometer and a contact microphone, a pressure-sensitive tablet for capturing ground truth intensity values, and machine learning algorithms for regression of scratch intensity on a 0–600 milliwatts (mW) power scale that can be mapped to a 0–10 continuous scale.

**Results** We evaluate the performance of our algorithms on 20 individuals using leave one subject out cross-validation and using data from 14 additional participants, we show that our algorithms achieve clinically-relevant discrimination of scratching intensity levels. By doing so, our device enables the quantification of the substantial variations in the interpretation of the 0–10 scale frequently utilized in patient self-reported clinical assessments.

**Conclusions** This work demonstrates that a finger-worn device can provide multi-dimensional, objective, real-time measures for the action of scratching.

## Plain language summary

Chronic itch can be caused by many medical conditions including eczema and psoriasis. Itch leads to scratching behaviors that can affect a person's sleep, productivity, mood, and overall well-being. We developed a ring device that can be placed on a person's finger to measure the intensity of scratching. Different types and intensities of scratching behavior could be distinguished in human volunteers. Further development of this device should enable more consistent and comprehensive measurement of scratching behaviors and help doctors and patients to better understand, and treat, chronic itch.

[1] Robotics Institute, Carnegie Mellon University, Forbes Avenue, Pittsburgh 15213 PA, USA. [2] Department of Dermatology, University of Pittsburgh Medical Center, Fifth Avenue, Pittsburgh 15213 PA, USA. [3] Mechanical Engineering, Carnegie Mellon University, Forbes Avenue, Pittsburgh 15213 PA, USA. [4]These authors contributed equally: Carmel Majidi, Zackory Erickson. ✉email: akhilpad@andrew.cmu.edu

Pruritus, commonly known as itch, is a debilitating symptom that can be caused by several conditions including atopic dermatitis, psoriasis, and liver disease, to name a few. Atopic dermatitis alone affects over 31 million people in the United States[1]. Pruritus leads to scratching behaviors, affecting the sleep, productivity, mood, and overall well-being of individuals[2–10]. The tracking of itch is important for determining treatment efficacy and enabling better patient outcomes. In practice, itch is quantified using uni-dimensional patient-reported scales such as the Visual Analog Scale (VAS), a 0–10 continuous measure, and the Numerical Rating Scale (NRS), a 0–10 discrete measure. While simple and quick to assess, these tools are subjective in nature and affected by anchoring and context bias, recall bias, and response shifts[11,12]. Moreover, they only provide a single data point at the time of measurement and thus are unable to provide a comprehensive understanding of the effect of the symptom on a patient. Multidimensional scales such as the 5-D Itch Scale and Itchy Quality of Life (ItchyQoL) provide more insight by tracking additional metrics such as degree (intensity of itching) and emotions, but suffer from the same downsides as the uni-dimensional scales and are additionally time consuming to assess[12–15].

Objective measures of pruritus focus on binary classification of scratching behaviors, which indicate whether a patient is scratching or not[16]. Tracking of these scratching behaviors provides insight into the effect of itch on an individual over time. Video recording using cameras (sometimes infrared) is the current standard in many studies[16,17]. However, this method is time intensive as it requires manual labeling of scratching events by researchers. This method can additionally suffer from blind spots, has privacy concerns, and is only possible while a patient is sleeping. Wearable devices have shown promise for automatic, objective classification of scratching at night[16,18–24]. Yet, advancements are needed to develop devices that perform well in daytime when participants still itch and when there are many confounding movements.

Furthermore, while existing wearable devices have the ability to record when a patient is scratching, the vast majority are unable to track scratch intensity. Scratch intensity can be related to itch intensity, which is tracked by doctors and patients using scales and questionnaires such as the Peak Pruritus NRS and the 5-D Itch Scale[13,25]. Scratch intensity is one of the factors that determines damage to skin and holds additional insight into the degree of disruption that itching has on everyday life and sleep. A wearable device, placed on the back of the hand, has previously been introduced to estimate scratching force during sleep[26]. In contrast to prior work, our system stands out due to its implementation of a multimodal sensing strategy with accelerometer and contact microphone data and our use of a pressure-sensitive tablet for training and validation, enabling us to attain statistically significant differentiation among various scratch intensity levels.

In this work, we introduce a framework for quantification of scratch intensity, a root cause of damage to skin due to chronic itch, and detection of scratching. Our framework consists of a wearable ring, shown in Fig. 1a, and multimodal algorithmic methods for real time analysis of scratch intensity and scratch detection. The multimodal ring consists of a 3-axis accelerometer and a vibration contact microphone, which provides complementary acousto-mechanic features on the scratching motion as shown in Fig. 1b. In conjunction, these sensors capture low frequency finger and arm motions, and high frequency vibrations propagating from the fingernail to the ring, as shown in Supplementary Movie 1. For each second of raw sensor data, our processing and prediction pipeline, shown in Fig. 2, extracts frequency features and uses two machine learning algorithms for

estimation of scratch intensity and detection in real-time. For training of our scratch intensity algorithm, we present methods for automatic generation of intensity labels using a touch sensitive surface embedded with an array of pressure sensors, i.e., a pressure-sensitive tablet. We assess the performance of our data-driven methods using leave one subject out (LOSO) cross-validation (CV) on data collected from 20 healthy participants, demonstrating the ability to accurately estimate scratch intensity on our proposed 0–600 mW power scale with a mean absolute error (MAE) of 49.71 mW. Moreover, we highlight the advantages of multimodal sensing for scratch detection, achieving 89.98% accuracy using the combined accelerometer and contact microphone model in comparison to 86.24 and 79.98% accuracy for the accelerometer only and contact microphone only models respectively. Finally, we extend the evaluation of our scratch intensity algorithm to include 14 new participants, achieving clinically relevant discrimination of intensity levels on a 0–10 scale with a MAE of 1.37 units.

## Methods

**Device design.** The ring device consists of dual complementary sensing modalities as shown in Fig. 1a: accelerations from an accelerometer and finger vibrations from a contact microphone. A contact microphone is a piezo-electric element that senses vibrations through solid objects (i.e., they are unaffected by acoustic waves traveling through the air, hence eliminating audio and speech privacy concerns). The ring was fabricated using the Sparkfun ADXL362 3-axis accelerometer breakout board, which measures accelerations between $\pm 2 \times g$, and PUI Audio Inc. AB1070B-LW100-R, which has a resonant frequency of 7 kHz. A custom printed circuit board (PCB) holds the PJRC Teensy 4.0 microcontroller and additionally has a non-inverting operational amplifier with a gain of 11 to amplify the signal from the contact microphone, a 1 MOhm resistor to remove drift from the contact microphone signal, and a protection diode for the Teensy. The Teensy microcontroller reads the amplified contact microphone signal using an 8-bit analog to digital converter (ADC). As a result, the recorded values range from 0 to 1023 (corresponding to 0–3.3 V).

For our experiments, the ring is worn on the dominant hand's index finger and secured using 3M Transpore medical grade tape and a compression finger sleeve. It is connected to the printed circuit board that can be worn elsewhere on the body, in this case, on the wrist using a velcro band and Coban self-adherent wrap. Data was sampled from the contact microphone and accelerometer at 8 kHz and 400 Hz, respectively.

## Scratch intensity

*Labeling.* As itch increases in severity, patients often scratch with a combination of increased normal force and velocity. We propose using mechanical power as a metric for scratch intensity, calculated by taking the product of force and velocity. Supervised machine learning algorithms can be used to obtain a mapping between wearable data and scratching power. It is non-trivial to obtain ground truth measurements of the power at which a person scratches (required for supervised learning) as force and velocity cannot be estimated accurately through human observation or techniques such as motion capture. As shown in Fig. 3, we instead generate power labels automatically by capturing scratching activity on a pressure-sensitive tablet (Sensel Morph[27]). The Sensel Morph has a sensing area of 30,960 mm² (240 mm by 139 mm) consisting of 19,425 pressure sensors (185 columns × 105 rows) at a 1.25 mm pitch (separating distance between sensors), with 5 g to 5 kg sensing range per touch. The tablet's overlay is removed and the device is instead layered with a

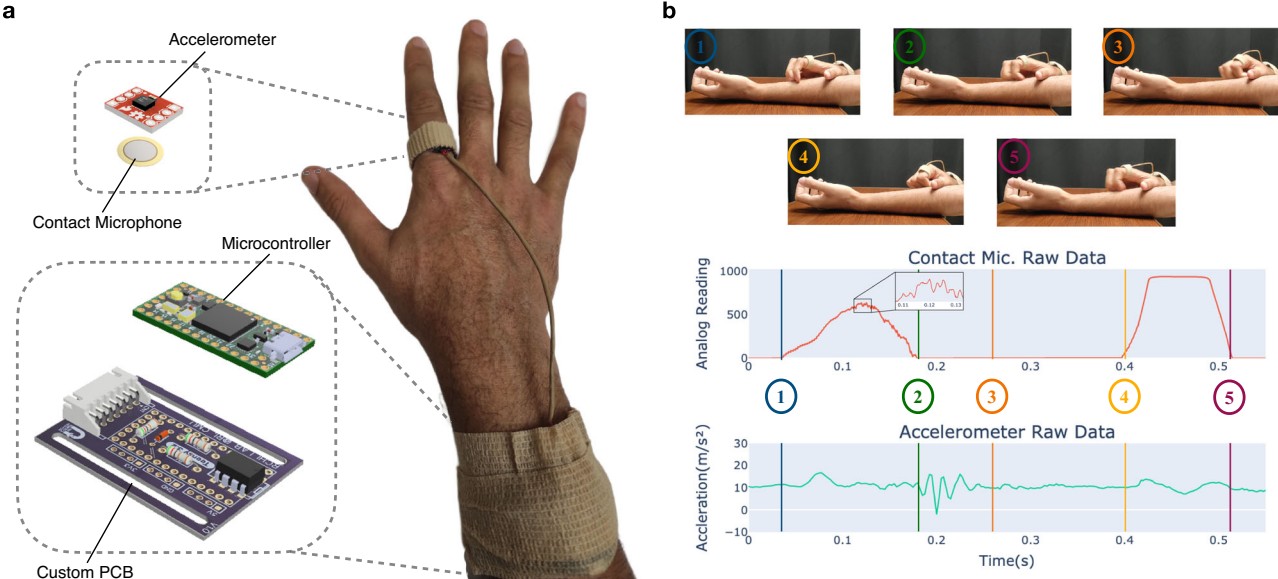

**Fig. 1 Proposed multimodal wearable ring and raw data during scratching. a** Components of the proposed multimodal wearable device. Right: human hand with wearable device donned. Top left: Rendering of the sensor suite inside of the ring consisting of an accelerometer and contact microphone. Bottom left: rendering of the custom printed circuit board with an operational amplifier for the contact microphone, a microcontroller, and other supporting electronics. **b** Clips from a finger movement only scratching motion are shown with corresponding raw data from the contact microphone and accelerometer (z-axis). 1–3: Fingers move up forearm. 4–5: Fingers move back down arm. We include Supplementary Movie 1, which shows synchronized contact microphone and wearable data for scratching behavior.

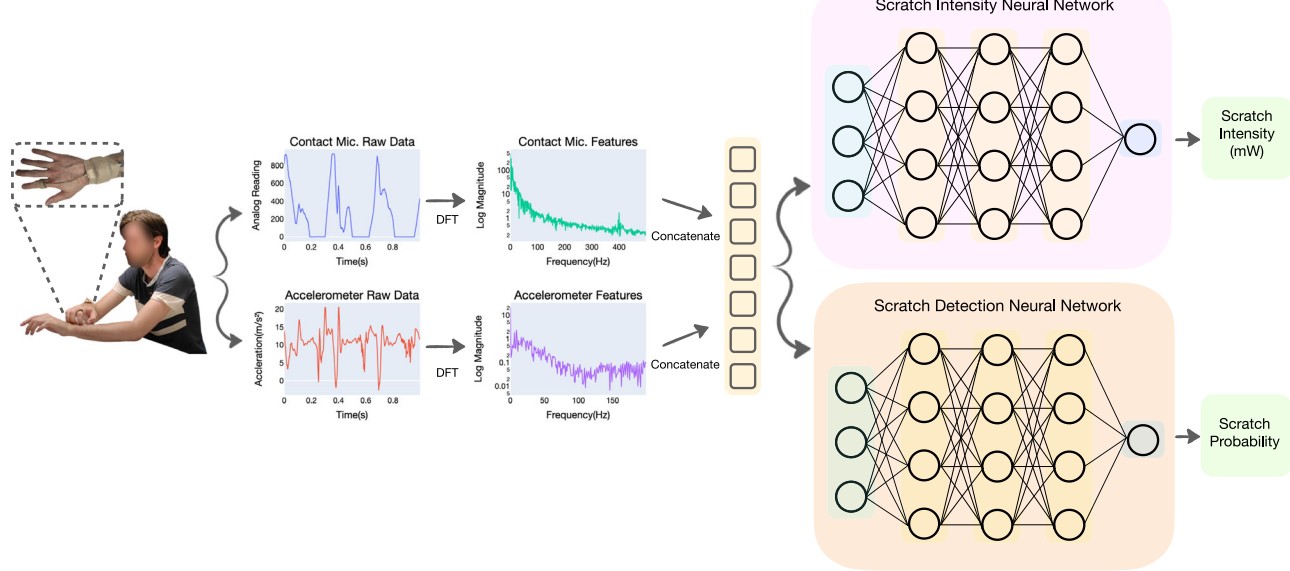

**Fig. 2 Scratch intensity and detection pipeline.** The Discrete Fourier Transform (DFT) is used to extract frequency features from the raw contact microphone and accelerometer data. The features are concatenated, normalized, and input into two neural networks, which independently regress to an estimate of scratch intensity and a scratch behavior probability. Permission was obtained from the participant for use of their image in publication.

100% cotton cloth which is fastened to the tablet to prevent creasing. The cloth is applied to the tablet as it has a texture more similar to human skin than the hard surface of the tablet and patients with pruritus may scratch over their clothing to satisfy an itch. For each 1-s window of measurements, we extract mean force and velocity estimates, allowing us to calculate the power label. Henceforth, we will refer to each 1-s window of data as a sample.

At each timestep, $t$, the Sensel Morph generates a force map using its array of pressure sensors as seen in Fig. 3. This image is automatically analyzed by the Sensel Morph's firmware to extract individual contact(s). For each contact, the Sensel Morph outputs the force, $f_t$, and the centroid of the contact, $(x_t, y_t)$. The x and y axes of the Sensel Morph are shown in Fig. 3. Aside from the index finger where the ring was worn, participants were instructed to keep their palm and fingers from touching the tablet during data collection to ensure there is only a single point of contact at each timestep. We sampled data from the tablet using the Sensel Morph API at 150 Hz. The entire labeling process is visualized in Fig. 3.

We create a force algorithm, shown visually in Fig. 3, to estimate mean force, $\bar{F}^s$, for each sample. Note that when no

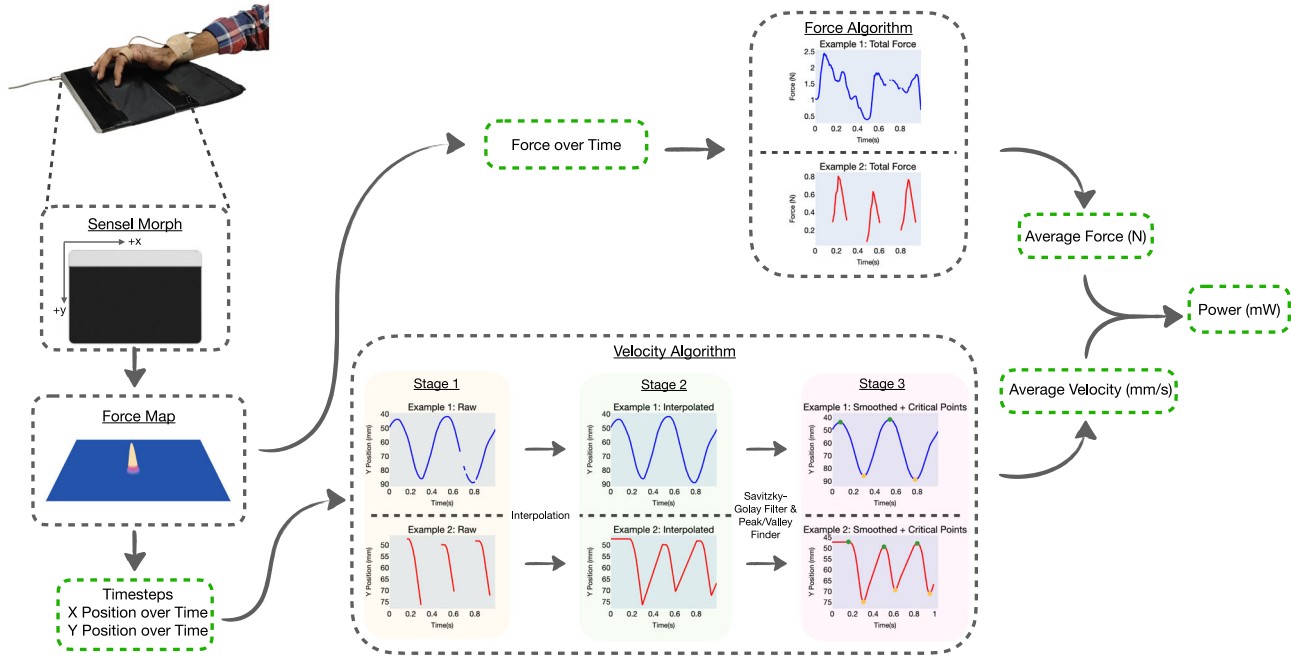

**Fig. 3 Scratch power labeling using a pressure-sensitive tablet.** The pressure-sensitive tablet outputs force, *x*-position, and y-position over time for each contact. This data is fed through our force and velocity algorithms to calculate average force and average velocity for each sample of data. Power is subsequently calculated by multiplying average force and velocity. Permission was obtained from Sensel Inc. for use of the Sensel Morph image.

contact is sensed by the Sensel Morph, it returns NaN; our method discards these values before calculating mean force using

$$\bar{F}^s = \frac{\sum_{j=1}^{n} F_j^s}{n} \quad (1)$$

where $F_j^s \in \boldsymbol{F}^s$ is the force value for a single timestep from the array of all force values $\boldsymbol{F}^s$ for a sample of data, and $n$ is the number of values in the sample.

Calculating average velocity is more involved. As visualized in the velocity algorithm section of Fig. 3, the majority of scratching behavior occurs in one of two ways; either the individual maintains contact with the skin through the entire scratching motion as shown in Example 1 or they lift up their finger off of their skin in between each scratching motion as shown in Example 2. In our human studies, discussed in the Results, we had participants scratch on the Sensel Morph for $z$ s at a time; as an example, in the first human study, we had participants scratch for $z = 10$ s. At a high level, velocity can be determined through the segmentation of $z$ s of data into distinct scratching motions. These motions are characterized by the finger moving up or down along the *y*-axis which are identified by detecting peaks and valleys (critical points) in the *y*-position signal from the Sensel Morph. By calculating the change in distance and the change in time during each of these individual scratching motions, we can obtain the average velocity across each 1-s sample. This process is explained mathematically as follows.

For every $z$ s of data with $n$ number of values, we have $\boldsymbol{x} = \left[x_1, x_2, \dots, x_n\right]$, the array consisting of the *x*-axis values of the finger contact over time, $\boldsymbol{y} = \left[y_1, y_2, \dots, y_n\right]$, the array consisting of the y-axis values of the finger contact over time, and $\boldsymbol{t} = \left[t_1, t_2, \dots, t_n\right]$, the timesteps corresponding to $\boldsymbol{x}$ and $\boldsymbol{y}$. In Fig. 1, Stage 1 shows 1 s of $\boldsymbol{y}$ vs $\boldsymbol{t}$ data for Example 1 and Example 2. First, we interpolate both $\boldsymbol{x}$ and $\boldsymbol{y}$ to fill in any gaps where contact is broken (i.e., values that are NaN) between the finger and tablet. This yields $\boldsymbol{x}^I \in \mathbb{R}^n$ and $\boldsymbol{y}^I \in \mathbb{R}^n$. Stage 2 shows 1 s of $\boldsymbol{y}^I$ vs $\boldsymbol{t}$ data for Example 1 and Example 2. Next, as shown in Stage 3, a fifth order polynomial Savitzky–Golay filter with a filter

window length of 0.21 s (31 points) is applied to smooth $\boldsymbol{y}^I$, yielding $\boldsymbol{y}^S$. These parameters were chosen through visual observation of the sensed contacts and generated peaks and valleys from a randomly selected subset of samples from each participant. For the $j$th point in $\boldsymbol{y}^S$, the formula for the Savitzky–Golay filtering is

$$y_j^S = \sum_{i=\frac{1-m}{2}}^{\frac{m-1}{2}} L_i y_{j+i}^I, \frac{m+1}{2} \le j \le n - \frac{m-1}{2}, \quad (2)$$

where $m = 31$ (the window size) and the convolution coefficients, $L_i$ can be looked up in the Savitzky–Golay tables. Also shown in Stage 3, a peak and valley finding algorithm[28] is applied to $\boldsymbol{y}_S$, identifying peaks and valleys (critical points) i.e., each incident at which the finger's velocity changes sign along the trajectory of the finger's scratching motion. Peaks are shown in green while valleys are shown in yellow. The vector of critical points is defined as $\boldsymbol{y}^C \in \mathbb{R}^m$, where $m$ is the number of critical points in the $z$ s of data. We additionally have $\boldsymbol{x}^C \in \mathbb{R}^m$, the corresponding $x$ positions for each critical point and $\boldsymbol{t}^C = \left[t_1^C, t_2^C, \dots, t_m^C\right] \in \mathbb{R}^m$, the corresponding timesteps for each critical point.

Next, in order to calculate a velocity estimate, $\bar{v}^s$, for each 1 s sample, we select a subset of the critical points $\boldsymbol{y}^{C,s} \in \mathbb{R}^q$, where $q$ is the number of critical points which occur in the 1-s window of time. We additionally select the corresponding $x$ positions, $\boldsymbol{x}^{C,s} \in \mathbb{R}^q$, and timesteps, $\boldsymbol{t}^{C,s} \in \mathbb{R}^q$. We calculate Euclidean distance (mm) between each adjacent pair of $\boldsymbol{x}^{C,s}$ and $\boldsymbol{y}^{C,s}$, forming an array of distances,

$$\boldsymbol{d}^{C,s} = \left[ \sqrt{\left(y_2^{C,s} - y_1^{C,s}\right)^2 + \left(x_2^{C,s} - x_1^{C,s}\right)^2}, \dots, \right.$$
$$\left. \sqrt{\left(y_q^{C,s} - y_{q-1}^{C,s}\right)^2 + \left(x_q^{C,s} - x_{q-1}^{C,s}\right)^2} \right] \in \mathbb{R}^{q-1}. \quad (3)$$

We can also find the difference in timesteps,

$$\Delta \boldsymbol{t}^{C,s} = \left[ t_2^{C,s} - t_1^{C,s}, t_3^{C,s} - t_2^{C,s}, \dots, t_q^{C,s} - t_{q-1}^{C,s} \right] \in \mathbb{R}^{q-1}. \quad (4)$$

We then calculate an array of velocities using

$$\boldsymbol{v}^s = \boldsymbol{d}^{C,s}/\boldsymbol{\Delta t}^{C,s}. \tag{5}$$

Lastly, we find the mean velocity for the 1-s sample,

$$\bar{v}^s = \sum_{a=1}^{q-1} v_a^s (q-1) \tag{6}$$

where $v_a^s \in \boldsymbol{v}^s$ is the velocity estimate between two adjacent critical points from the array of all velocities $\boldsymbol{v}^s$ for a sample of data. For each sample, the power label in mW, $p^s$, is found using

$$p^s = \bar{F}^s \bar{v}^s, \tag{7}$$

where $\bar{F}^s$ (N) is the calculated average force and $\bar{v}^s$ (mm/s) is the average velocity for the sample.

To mitigate errors in the power labeling procedure above, we remove outliers in our velocity algorithm. We filter using 3 conditions: 1. If there are less than 2 peaks/valleys in a 1-s window of data. 2. If there are consecutive peaks or consecutive valleys detected in a window of data. 3. If the change in contact $x$ or $y$ positions in consecutive timesteps is greater than 5 mm. Lastly, we remove any outliers greater than 600 mW as power values higher than this limit are often infeasible.

*Dataset, feature extraction, and model design.* For each 1-s sample of wearable data, we use linear interpolation to fill in any missing values in the contact microphone and accelerometer $z$-axis (normal to skin) signals, $\boldsymbol{c}$ and $\boldsymbol{a}$ respectively. Thus, we have $\boldsymbol{c}^I \in \mathbb{R}^{8000}$ and $\boldsymbol{a}^I \in \mathbb{R}^{400}$ as two interpolated signals in the time domain. Power information is captured in the frequency domain of the contact microphone and accelerometer data. We explore this idea further in the Results using participant data from a human study. The Discrete Fourier Transform (DFT), $\mathcal{F}$, can be applied to a discrete signal in the time domain to convert it to the frequency domain. For the $k$th point in the DFT result, $\boldsymbol{d}$, the DFT formula is

$$\boldsymbol{d}_k = \sum_{n=0}^{N-1} e^{-2\pi i \frac{kn}{N}} \boldsymbol{g}_n \tag{8}$$

where $\boldsymbol{g}$ is the time domain signal and $N$ is the length of $\boldsymbol{g}$. We apply DFT and extract the single-sided frequency amplitude for the contact microphone signal,

$$\boldsymbol{c}^A = \frac{2}{B} \mathcal{F}(\boldsymbol{c}^I), \tag{9}$$

where $B = 8000$. Similarly, for the accelerometer,

$$\boldsymbol{a}^A = \frac{2}{E} \mathcal{F}(\boldsymbol{a}^I), \tag{10}$$

where $E = 400$.

From $\boldsymbol{c}^A$, we select the first 400 amplitudes, yielding contact microphone features, $\boldsymbol{c}^G \in \mathbb{R}^{400}$, which corresponds to frequencies from 0 to 399 Hz. From $\boldsymbol{a}^A$, we select the first 175 amplitudes yielding accelerometer features, $\boldsymbol{a}^G \in \mathbb{R}^{175}$, which corresponds to frequencies from 0 to 174 Hz. These features are concatenated, forming a feature vector, $\boldsymbol{G} \in \mathbb{R}^{575}$. Min–max normalization, fitted on the training data, was applied to $\boldsymbol{G}$, scaling all features to between 0 and 1. This processing pipeline is shown visually in Fig. 2.

Using our normalized wearable features and power labels from the Sensel Morph, we trained a neural network for regression of scratch intensity. To find the best model, we conducted a grid search over model hyperparameters. The hyperparameters that we searched over were the number of hidden layers, the number of nodes in the hidden layers, dropout probability, and the number of accelerometer and contact microphone frequency features. For each model, we used LOSO-CV to calculate the

MAE across all folds and the model with the lowest MAE was picked. A fully connected neural network with 2 hidden layers and hidden layer size of 1000 nodes with ReLU activation had the best performance. During training, we applied a dropout ($p = 0.1$) to each hidden layer to reduce overfitting. Lastly, we trained using the Adam optimizer with a learning rate of $5*10^{-6}$, MAE loss, batch sizes of 64, and 150 epochs. Train/Test plots are shown in Supplementary Fig. S1.

**Scratch detection experimental design**
*Dataset, feature extraction, and model design.* Supervised machine learning in conjunction with frequency features from the proposed multimodal wearable ring can additionally be used for detection of scratching behavior. Wearable data is divided into 1-s samples using a stride of 0.25 s and features were extracted using the DFT. From $\boldsymbol{c}^A$, the first 275 amplitudes are selected yielding contact microphone features, $\boldsymbol{c}^H \in \mathbb{R}^{275}$, which corresponds to frequencies from 0 to 274 Hz. From $\boldsymbol{a}^A$, the first 200 amplitudes are selected yielding accelerometer features, $\boldsymbol{a}^H \in \mathbb{R}^{200}$, which corresponds to frequencies from 0 to 199 Hz. These features are concatenated, forming a feature vector, $\boldsymbol{H} \in \mathbb{R}^{475}$. Min-max normalization, fitted on the training data, was applied to $\boldsymbol{H}$, scaling all features to between 0 and 1.

The normalized frequency features are used to train a fully connected neural network with 3 hidden layers with ReLU activations and a hidden layer size of 1200 nodes. A sigmoid activation was applied to the output layer. During training, we applied dropout ($p = 0.2$) to each hidden layer. We used binary cross-entropy loss and the Adam optimizer with a learning rate of $1*10^{-5}$. The model was trained for 150 epochs with batch sizes of 64. Similarly to the intensity regression model, we found the best scratch detection model by conducting a grid search over model hyperparameters. The hyperparameters that we searched over were the number of hidden layers, the number of nodes in the hidden layers, dropout probability, and the number of accelerometer and contact microphone frequency features. Train/test plots are shown in Supplementary Fig. S1.

**Statistics and reproducibility.** For evaluation of all of our models, we use LOSO-CV. In LOSO-CV, $n$ iterations (folds) of CV are conducted where $n$ is the number of participants. For each iteration, a single participant's data is held out as a test set while the model is trained on all other participant data. We utilize LOSO-CV as it mirrors the clinically relevant scenario of tracking scratching in new subjects whose data is not in the training set[29]. Metrics are averaged over all held-out participants (all test folds from LOSO-CV).

For intensity regression, we use MAE as a metric and as our loss function. For an array of ground truth labels, $\boldsymbol{y} \in \mathbb{R}^n$, and an array of predictions, $\hat{\boldsymbol{y}} \in \mathbb{R}^n$, MAE is calculated using the following formula:

$$\text{MAE(mW)} = \frac{\sum_{i=1}^n |y_i - \hat{y}_i|}{n}. \tag{11}$$

We additionally use mean absolute percentage error (MAPE) as a metric for better interpretability of our results. MAPE is calculated using the following equation:

$$\text{MAPE(\%)} = \frac{\sum_{i=1}^n |y_i - \hat{y}_i|}{600n} \tag{12}$$

as 600 mW is the max power label on our 0–600 mW power scale. For detection of scratching behavior, we use mean classification accuracy as a metric and use binary cross entropy loss for our loss function.

For additional evaluation of our intensity regression results, we use the Wilcoxon signed-rank test and Spearman's rank correlation coefficient. Further details on these statistical tests are located in the Results section.

For reproducibility of our results, we release all data and code as detailed in the "Data availability" and "Code availability" sections. Underlying data for all figures can be found in Supplementary Data 1–10.

**Reporting summary**. Further information on research design is available in the Nature Portfolio Reporting Summary linked to this article.

## Results

We conducted a human study (Carnegie Mellon University IRB Approval under 2021_00000480) with 24 healthy participants, of which data from 20 participants was used. Three participants' data was disregarded due to sensor failures during data collection while one participant was dis-enrolled prior to data collection due to not meeting inclusion criteria. Summarized demographics information can be found in Fig. 4a. Written informed consent was obtained from all participants. The participants were affixed with the wearable ring and instructed to scratch with just their index finger contacting the tablet, allowing our labeling algorithm to estimate power. They were led through the 9 combinations of low, medium, and high force, and low, medium, and high speed scratching on the Sensel Morph for 10 s each. The ordering of the combinations was low force low speed, low force medium speed, low force high speed, medium force low speed, medium force medium speed, medium force high speed, high force low speed, high force medium speed, and high force high speed. Participants have differing interpretations of low, medium, and high force and speed which allows us to collect a diverse dataset of varying scratching behavior with a large range of power labels.

Dataset information is shown in Fig. 4b. As specified in the "Methods," we split the data into 1-s windows of sensor measurements with an associated power label automatically generated via the Sensel Morph pressure tablet. We capture a total of 30 min of scratching behavior across all 20 participants (approximately 1.5 min of scratching per participant) resulting in 6600 data samples as seen in Fig. 4b. After cleaning erroneous labels, there are 4227 samples. From this cleaned dataset, estimated force measurements are grouped for each instruction of low, medium, and high force and are shown on the left side of Fig. 4d. The mean and standard deviation for forces are $0.36 \pm 0.23$, $0.66 \pm 0.29$, and $1.56 \pm 0.58$ Newtons for instructed low, medium, and high force, respectively. Similarly, estimated velocity measurements from the Sensel Morph are shown on the right side of Fig. 4d for each instruction. The mean and standard deviation for velocities are $111.97 \pm 48.74$, $136.36 \pm 56.04$, and $177.93 \pm 62.67$ mm/s$^2$ for instructed low, medium, and high velocity respectively. The power labels are shown in Fig. 4e for each combination of low, medium, and high force, and low, medium, and high speed and range from 0 to 600 mW. The dataset is right skewed, with the vast majority of labels in the 0–200 mW range, seen by the histogram of all power labels in Fig. 4f.

We observe that power information is captured in the frequency domain of the contact microphone and accelerometer z-axis (normal to skin surface) data, which is transformed from the time domain using the Discrete Fourier Transform (DFT). As force and velocity increase, the amplitude of the vibrations between the fingernail and skin correspondingly rise. In Fig. 4c, each line corresponds to one of the nine combinations of low, medium, and high force and low, medium, and high speed that participants were instructed to perform. For each of these combinations, we transform each sample into the frequency domain, min–max normalize to scale values to between 0 and 1, average all the signals across all the participants, and then apply smoothing using a Savitzky–Golay filter for visualization purposes. In Fig. 4c, note that as the instructed force changes from low, medium, to high, the magnitudes of the signals also increase for most frequencies. Similarly, as the instructed velocity changes from low, medium, to high, the magnitudes of the signals generally increase. Noticeably, in Fig. 4c, the vertical ordering of the signals in the higher frequency ranges (≥150 Hz for the contact microphone and ≥70 Hz for the accelerometer and marked in red in the plots) directly matches the ordering of the power values recorded by the tablet, shown in Fig. 4e. These results suggest that frequency domain features from wearable data can be used for regression of scratching power. As discussed in the Methods, for the contact microphone data, we specifically use the first 400 amplitudes, which corresponds to frequencies from 0 to 399 Hz while for the accelerometer data, we use the first 175 amplitudes yielding accelerometer features, which corresponds to frequencies from 0 to 174 Hz.

**Scratch intensity model performance**. As discussed in "Methods," normalized frequency features in conjunction with ground truth power labels from the pressure-sensitive tablet are used to train a neural network for regression of scratch intensity. We utilize LOSO-CV for evaluating model architectures and for tuning model parameters. Performance metrics, specifically MAE and MAPE, are then averaged across all folds of CV.

Figure 4g presents results for scratch intensity regression using LOSO-CV across the 20 healthy participants from the first human study. The model trained using both accelerometer and contact microphone features performs the best with a MAE of 49.71 mW and a MAPE of 8.29%. The minimum MAPE for 1 held-out participant (1 fold) is 2.33% while the maximum MAPE is 17.20%. Detailed metrics for each fold are shown in Supplementary Table S1. As shown in Fig. 4g, we conduct an ablation study. For this ablation study, we evaluate our algorithm with just the contact microphone features and just the accelerometer features to compare the performance of single-sensor approaches with the multimodal model. The model trained using just accelerometer features performs similarly with a MAE of 50.58 mW and a MAPE of 8.43% while the model trained only with contact microphone features has a MAE of 60.66 mW and a MAPE of 10.11%. All models perform better than the naive predictor which uses the mean, 119.64 mW, of all the labels as the prediction. Figure 4h shows the spread of test errors from LOSO-CV for different power label ranges.

**Scratch intensity validation**. We trained a scratch intensity model using all 20 participants' data from the first human study with optimal model hyperparameters found using a grid search. We then validated the performance of this trained model through a second human study (Carnegie Mellon University IRB Approval under 2021_00000480) in which 14 healthy participants were instructed to perform various intensities of scratching. Demographics information for this study can be found in Fig. 5a. Written informed consent was obtained from all participants.

Each participant participated in 4 sets of instructed scratching. Each set consists of scratching at intensities from 1 to 5 with 8 s of scratching at each intensity. Participants were told that 1 is the least intensely they would scratch on their skin to satisfy an itch while 5 is the most intensely they would scratch on their skin to satisfy an itch. Participants conducted the first two of these sets on the pressure-sensitive tablet, scratching with just their index

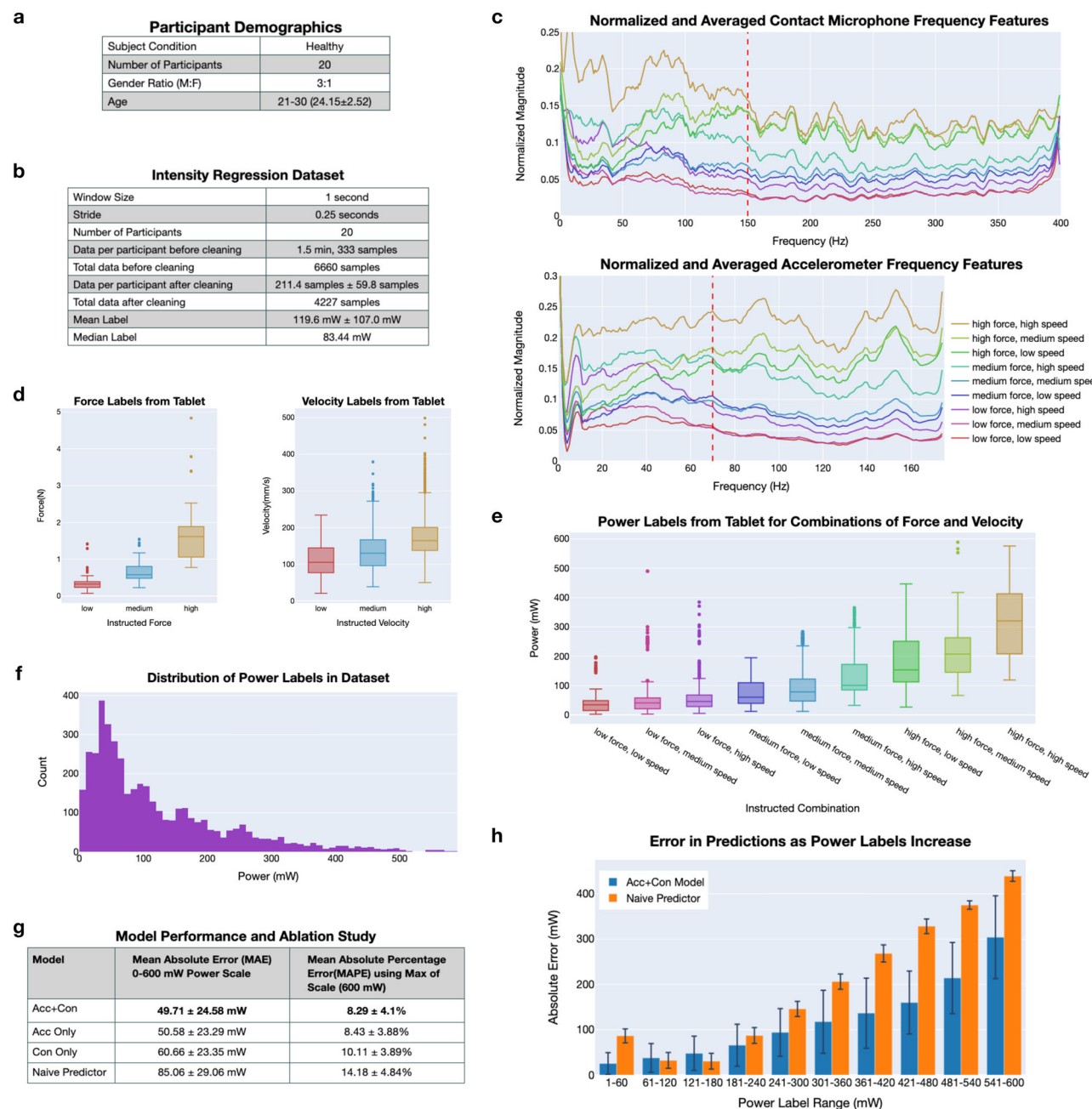

**Fig. 4 Scratch intensity dataset, features, and performance using leave one subject out cross validation. a** Participant demographics from first human study. **b** Details on the intensity regression dataset. **c** Contact microphone and accelerometer DFT features for combinations of force and velocity are min-max normalized to scale values to between 0 and 1, averaged across all participant samples, and subsequently smoothed using a Savitzky–Golay filter for visualization. Vertical ordering of the DFT amplitudes greater than 150 and 70 Hz (designated by the red dashed vertical lines) match the ordering of power labels in (**e**). **d** Box plots of the recorded forces and velocities from the Sensel Morph as participants scratched on the tablet at combinations of varying force and velocities. **e** Box plot of power labels from all participant data for combinations of force and velocity. **f** Histogram of all power labels (n = 4227) in the dataset. **g** Performance using LOSO-CV and ablation study over sensor features used. ± denotes standard deviation. **h** Box plot of errors in predictions on test data for various ranges of labels. The naive predictor uses the mean of all labels as the prediction. Error bars indicate 1 standard deviation from the mean. Underlying data for this figure can be found in Supplementary Data 1–4.

finger to allow us to record ground truth power estimates using the Sensel Morph. Participants were then instructed to conduct two more sets on their skin, each at a location of their choosing and participants were allowed to scratch normally with no restrictions on the fingers used. All participants chose to scratch on either their medial forearm, lateral forearm, medial upper arm, or lateral upper arm. Sensel Morph data was labeled and wearable data was processed using the same pre-processing steps as used for the training data.

Using data from the two sets of scratching on the pressure-sensitive tablet, Fig. 5b presents results for intensity regression for the 14 healthy participants from the second human study. Our multimodal wearable and trained intensity regression model achieves a MAE of 57.4 mW and a MAPE of 9.57%. The minimum MAPE for 1 held-out participant (1 fold) is 2.92% while the maximum MAPE is 24.15%. Detailed metrics for each fold are shown in Supplementary Table S2 and errors for different power label ranges are shown in Supplementary Fig. S2.

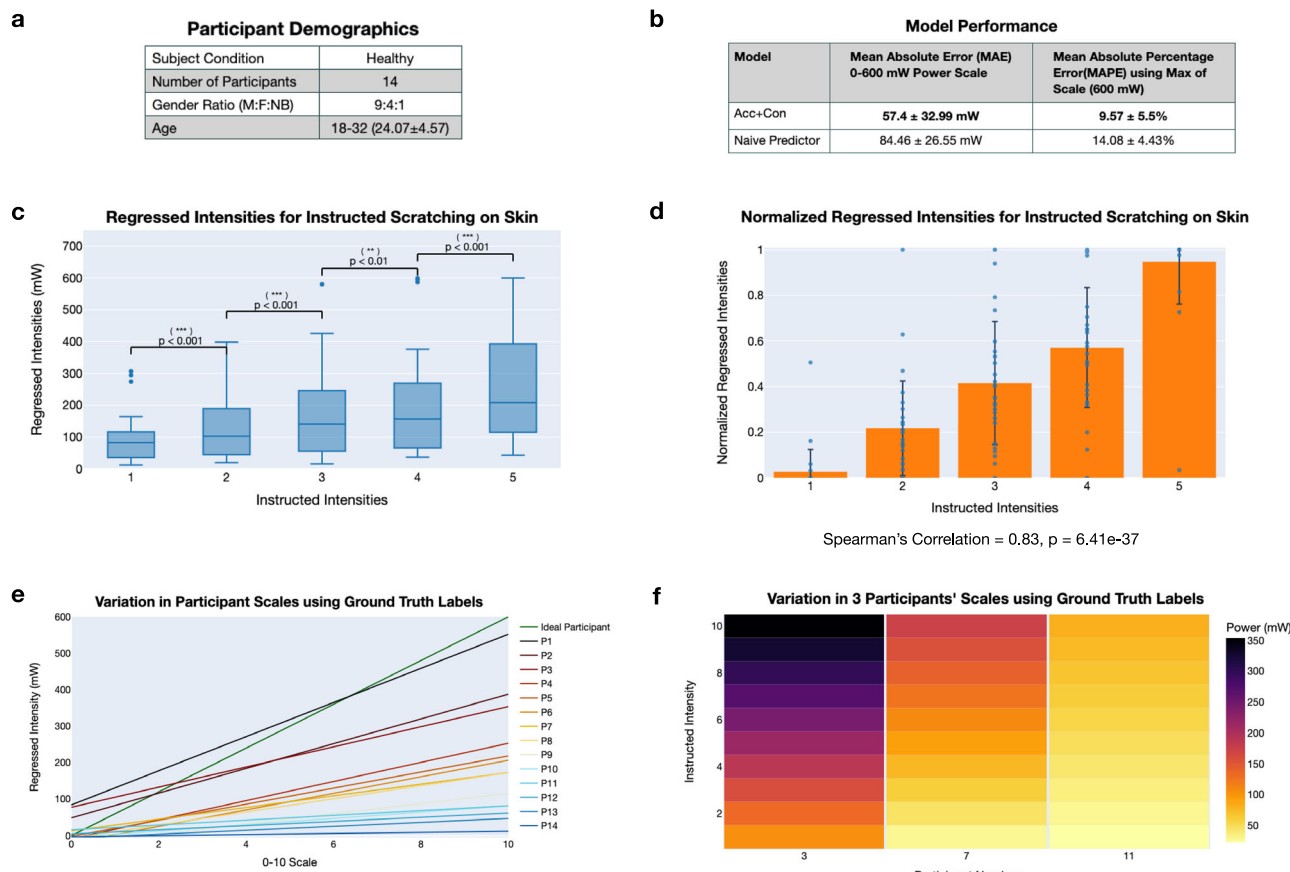

**Fig. 5 Scratch intensity validation performance. a** Participant demographics for second human study. **b** Model performance on new scratch intensity data. ± denotes standard deviation. **c** Regressed intensities on the 0–600 mW power scale for scratching on skin for instructed intensities from 1 to 5. Regressed intensities are grouped based on the instructed intensity. For each instructed intensity, $n = 28$. Statistically significant differences are found between all groups based on a Wilcoxon signed-rank test. **d** Normalized regressed intensities for scratching on skin grouped by the instructed intensity value. A Spearman's correlation of 0.83 is calculated showing a strong positive correlation. Bars indicate the mean and error bars indicate 1 standard deviation from the mean. For each instructed intensity, $n = 28$. **e** The variation in individually fit participant intensity scales using ground truth labels from the pressure-sensitive tablet is shown. For each participant, a line is fit to the data from both sets of scratching on the Sensel Morph. **f** Variation in three participants' scales using ground truth labels from the pressure-sensitive tablet is shown. Underlying data for this figure can be found in Supplementary Data 5–8.

Using data from the two sets of scratching on skin, we show that our learned model can differentiate between scratching intensity levels on skin. For all participants, for each instructed intensity, we calculate the mean regressed intensity value across the 8 s of scratching on skin. Figure 5c shows that as the instructed intensity increases, the mean regressed intensity likewise increases from 96.52 to 129.17 to 170.92 to 203.78 to 262.03 mW. Using the Wilcoxon signed-rank test, we find statistical significance between each intensity level and especially strong significance between intensity levels 1 and 2 and intensity levels 4 and 5. For each intensity level, $n = 28$, as each of the 14 participants scratches twice at each intensity. The exact $p$ values are 1.68e−5, 3.50e−4, 1.55e−3, and 2.79e−5. Figure 5d additionally shows our device's ability to differentiate between scratching intensities on skin. Similarly to the last figure, for all participants, for each instructed intensity, we calculate the mean regressed intensity value across the 8 s of scratching on skin. To account for different understandings of the 1–5 scale by participants and variation in the two sets, we apply min–max normalization across the averaged regressed intensity values from each set. In Fig. 5d, we show the mean and standard deviation across all participants for the normalized sets and additionally overlay all normalized data points. The means for intensity levels 1–5 are 0.027, 0.218, 0.416, 0.572, and 0.948, respectively. The

Spearman's rank correlation coefficient for this data is 0.83 with a $p$ value of 6.41e−37 implying a strong positive correlation in the data.

Our trained model predicts the power (in mW) associated with a scratching action. However, the 0–600 mW scale does not hold much interpretability to patients or their doctors and thus we present a method for converting this scale to a 0–10 continuous scale (similar to the VAS commonly used in dermatology assessment). We apply a simple linear scaling by dividing all labels and predictions by 60. Thus, the bounds in power units for the 0–10 scale are the following: 0, 60, 120, 180 ... 540, 600 mW. In the new 0–10 scale, after the scaling, the MAE (from Fig. 5b) becomes 0.96 units with a standard deviation of 0.55 units. In Supplementary Fig. S3, we additionally present an alternate nonlinear square root function which achieves a MAE of 1.37 units.

Through capturing ground truth power labels with the pressure-sensitive tablet, we observe that people have very different interpretations of the 0–10 continuous scale. For example, a scratch intensity of 6 for one person is an intensity of 1 for another person. To demonstrate this, using ground truth values from the two sets of scratching on the Sensel Morph, for each participant, for each instructed intensity from 1 to 5, we linearly scale the ground truth values to the 0–10 scale. For each participant, we fit a line to this data as shown in Fig. 5e. The lines

have an average slope of 18.22 mW/unit (sd: 12.24) and an average $y$-intercept of 12.17 mW (sd: 32.93), showing a large difference in scales for each participant. The ideal line, shown in dark green in Fig. 5e, has a $y$-intercept of 0 as an intensity of 0 on the 0–10 continuous scale corresponds to no scratching (0 mW). It also has a slope of 60 as an intensity of 10 on the 0–10 continuous scale corresponds to 600 mW on the power scale. This ideal line is only possible with a "perfect" participant who understands the power scale and is able to control their scratching intensity exceptionally well. Figure 5f further shows the difference in scales between 3 participants using a heatmap. Participant 3's scale ranges from 105.64 mW (corresponding to 1 on their subjective version of the 0–10 scale) to 354.00 mW (corresponding to 10 on their subjective version of 0–10 scale). Participant 7's scale ranges from 29.43 to 173.37 mW. Lastly, Participant 11's scale ranges from 23.01 to 81.95 mW.

**Scratch detection dataset and feature extraction**. To develop and validate our algorithms for scratch detection, we collected a dataset from 20 healthy participants. Demographics are shown in Fig. 6c. Participants were instructed to scratch on different locations of their body (top of hand/fingers, forearm/wrist, inside elbow, neck, head, behind the knees, ankles) and to do a variety of non-scratching activities (hand waving, keyboard typing, texting/phone swiping, writing, table tapping, air scratching, clapping) for 30 s each. Raw data from the contact microphone and accelerometer for each scratching and non-scratching interaction made by a single representative participant can be seen in Fig. 6a, b. All data collected was divided into 1 s samples using a stride of 0.25 s. In total, there is 140 min of data with 32,760 samples. The dataset is balanced with equal number of scratching and non-scratching data. Dataset information can be found in Fig. 6b. As discussed in the Methods, we utilize the DFT to extract frequency features from wearable data and use these to train a neural network to detect scratching behavior.

**Scratch detection model performance**. In Fig. 6d, we present results for scratch detection using LOSO-CV across the 20 healthy participants. The model using both accelerometer and contact microphone features achieves an accuracy of 89.98% on the balanced dataset. The minimum accuracy for 1 held-out participant (1 fold) is 80.22% while the maximum accuracy is 98.05%. Detailed metrics for each fold are shown in Supplementary Table S3. Figure 6d also provides results from an ablation study to quantify the benefits of the multimodal sensing ring; classification accuracy drops from 89.98 to 86.24 and 79.98% with the exclusion of either the accelerometer and contact microphone, respectively.

Figure 6e provides the classification accuracy for each interaction. For the non-scratching interactions, the model performs especially well for hand waving and clapping, classifying them with 97 and 98% accuracy, respectively. It performs worst on the writing and tapping interactions, classifying them with 75 and 79% accuracy, respectively. For the scratching interactions, the model correctly classifies scratching behavior at all scratching locations with over 89% accuracy.

## Discussion

This work introduces a framework for the quantification of scratch intensity that included a multimodal wearable ring, methods for capturing ground truth intensity of scratching interactions from a pressure-sensitive tablet, and a supervised machine learning algorithm for regression of scratching intensity on a 0–600 mW power scale. We show how extracted frequency features from contact microphone and accelerometer raw data hold valuable information that allows for the regression of

scratching power using a neural network. We additionally quantify the performance of this method using LOSO-CV and showed it performs well in both human studies, including the validation study with data from 14 new participants that our models were not trained on.

We showed that our scratch intensity model, trained on data collected from scratching on a tablet, accurately captures scratching on the skin. Although we have a pressure-sensitive tablet for controlled validations, as a community, we lack the skin-tight pressure sensor array technology that would be necessary to quantify the error (in mW power units) that our method exhibits for on-skin scratching. Thus, to evaluate our algorithms, we instructed participants to scratch at intensities from 1 to 5 on their skin. Our model achieves statistical significance between the different intensities and additionally shows a strong Spearman's correlation after normalization, providing compelling evidence for the ability of a ring wearable and learned regression model to differentiate between scratching intensities on skin. The ability of our model to infer intensities on a 0–10 scale also maps to existing scales that are common in dermatology practice for evaluation of itch and scratching.

Through our human studies, we observed some outliers in performance; for example, there was low correlation for one participant between instructed intensities and regressed values for scratching on skin. Our analysis from data from both studies also found higher errors in predictions when scratching behaviors reached high power magnitudes. We suspect this is due to fewer samples in those higher bounds and due to some saturation in accelerometer and contact microphone signals.

Our method allows the research and dermatology communities to quantify for the first time the large differences in how people interpret the 0–10 scales that are commonly employed in clinical patient-reported assessments. The difference in participants' interpreted scratching behaviors in Fig. 5 and the 3 participants presented in Fig. 5f demonstrate the subjectivity of the 0–10 scale that is commonly employed in clinical practice. These results further motivate the need for a wearable device like the one presented that provides objective measures of scratch intensity.

We foresee intensity estimates from our device being utilized in various ways. The proposed power scale for scratch intensity holds physical meaning, allowing for future exploration of its relationship to skin damage. In clinical and drug trials, the regressed power values can be used directly by researchers for quantitative and statistical analysis of treatment efficacy. However, in outpatient clinical settings, this power scale holds little value and interpretability to practising doctors and patients, who are more familiar with existing scales such as the 0–10 continuous VAS or 0–10 discrete NRS. The data from the power scale can be converted to an objective 0–10 continuous scale, by applying a simple linear scaling or non-linear transformation on the output of the neural network, or a 0–10 discrete scale, by binning the labels and using classification algorithms. To make our results more understandable, we provide two examples of this. Using the 0–10 objective, continuous scale and a linear scaling, our method yields a MAE of 0.96 units and with a non-linear scaling, our method yields a MAE of 1.37 units. For our data, the linear model is superior to the non-linear model presented but for higher recorded scratching power, a non-linear model could fit the data better. For reference, in clinical trials, studies often use a change of 4 units on a subjective, patient reported version of the 0–10 scale as the minimal clinically important difference[30,31]. This suggests that a linear mapping of our power measurements to a 0–10 scale has sufficiently low error and would allow for clinically-relevant discrimination of scratching intensity levels. Additionally, in contrast to existing patient-reported scales which only capture the symptom of itch at a snapshot in time, this

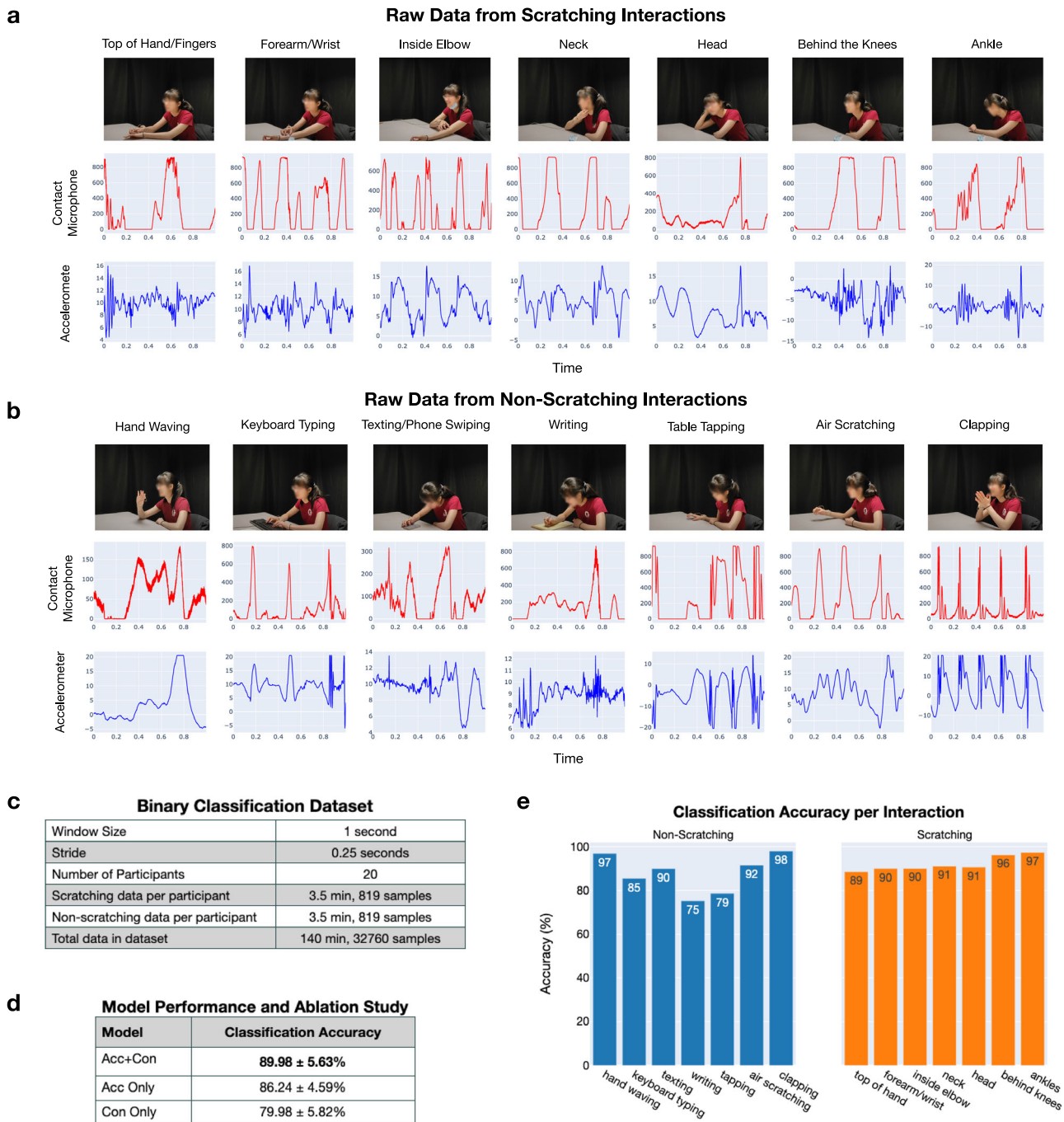

**Fig. 6 Scratch detection dataset and performance using leave one subject out cross validation. a** The 7 scratching interactions are shown with accompanying raw data from the contact microphone and accelerometer for a representative participant. **b** The 7 non-scratching interactions are shown with accompanying raw data from the contact microphone and accelerometer. **c** Details on the scratch detection dataset. **d** Results from the ablation study conducted using LOSO-CV. Acc stands for Accelerometer and Con stands for Contact Microphone. ± denotes standard deviation. **e** Classification test accuracy averaged across all folds for each interaction in the dataset using the best classification model (Acc+Con). Underlying data for this figure can be found in Supplementary Data 9. Permission was obtained from the participant for use of their image in publication.

technology has the potential to provide continuous monitoring without any added effort by doctors and patients.

Our results additionally show promise for improving binary classification for scratch detection. Using LOSO-CV, our model achieved 89.95% accuracy on the balanced dataset with equal numbers of scratching and non-scratching data. Our ablation results show that the proposed combination of two acousto-mechanic sensors yields superior performance over single sensor methods. This was especially the case for scratch detection, where the synthesis of

features resulted in an increase of 3.74 and 10.00% in performance over the accelerometer and contact microphone single sensor approaches, respectively. These results motivate further work in multimodal sensing wearables for detection of scratching behavior specifically for daytime use when there are confounding movements. While our study focused on differentiating scratching behaviors from many common daily activities, future work could additionally consider behaviors that may be prevalent or similar to scratching, such as kneading, picking, and rubbing.

The form factor and location of the ring wearable also presents benefits to the long-term adoption of the technology, especially for daytime use. Conspicuous, non-esthetic, assistive, and medical devices with symbolism of impairment have shown to be less positively accepted due to social stigmas[32,33]. While our prototype device is currently tethered and requires a battery and supporting electronics on the wrist in addition to the sensorized ring, there are existing, well-adopted commercial ring devices, such as the Oura Ring[34], which show that electronics with multiple sensors, including accelerometers, can be successfully embedded in a ring form factor with a long lasting battery life and esthetic design[35,36]. One limitation of this design choice is that the device only has the ability to track scratching motions by a sole finger on a single hand, which may result in undetected scratches. We foresee patients wearing the ring on the finger they scratch with the most regularly.

Another limitation of this work is the lack of evaluation with patients with pruritus. Future work will focus on development of a fully integrated, wireless ring for continuous scratch monitoring allowing long term real world studies that evaluate our algorithms in practice on representative patients, who may scratch differently than healthy participants. Additional testing and evaluation of our multimodal wearable ring for tracking scratching behaviors would be helpful before long-term deployment, including a signal strength and noise analysis for when the ring is placed at different locations along the index finger or moved to other fingers.

As proposed in past works, we envision this device automatically uploading this data to patient records in the cloud, where it could be subsequently aggregated, analyzed, and visualized in a mobile application[19,37,38]. This data can additionally be synced with other information such as sleep data, temperature/humidity, and patient-reported details such as diet, allowing individuals to identify correlations between potential itch triggers and their subsequent itch symptoms.

In conclusion, our work highlights a multimodal sensing ring for quantification of scratch intensity in addition to scratch detection, two essential metrics for the tracking of itch. Within clinical trials, this wearable and framework could provide researchers and clinicians with greater insight into treatment effectiveness via real-time signals that directly correspond to scratch behaviors. By providing direct objective quantification of scratching, our ring device has the potential to improve patient outcomes and aid with treatment discovery and validation.

## Data availability

The datasets collected, generated, and analyzed in the study are located at https://doi.org/10.5281/zenodo.8195949[39] and additionally on GitHub via this link: https://github.com/RCHI-Lab/Wearable_Scratch_Intensity. The data displayed in Fig. 4d–f, h is located in Supplementary Data 1–4, respectively. The data displayed in Fig. 5c–f is located in Supplementary Data 5–8, respectively. The data displayed in Fig. 6e is located in Supplementary Data 9.

## Code availability

The underlying code for this study, for the statistical analysis, and for generation of figures is located at https://doi.org/10.5281/zenodo.8195949[39] and additionally on GitHub via this link: https://github.com/RCHI-Lab/Wearable_Scratch_Intensity.

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

## Acknowledgements

This study was funded by the National Science Foundation Graduate Research Fellowship Program under Grant No. DGE1745016 and DGE2140739. Any opinions, findings, and conclusions or recommendations expressed in this material are those of the author(s) and do not necessarily reflect the views of the National Science Foundation. The funder played no role in study design, data collection, analysis and interpretation of data, or the writing of this manuscript. We would like to thank Jashkumar Diyora, Kadri Bugra Ozutemiz, and Anthony Wertz for assistance and advice with the custom PCB design and electronics. We would like to thank Tejus Gupta for insightful conversations.

## Author contributions

A.P., C.M., and Z.E. formulated the research idea and goals. A.P. designed, built, and programmed the wearable device and wrote the data collection scripts. A.P. developed and validated the classification and regression algorithms. A.P. wrote the IRB protocol, collected data during both human studies, and analyzed the data. S.C. provided advice on the medical implications of the work and study design. A.P. created the figures and wrote the paper draft. All authors edited the paper. C.M. and Z.E. were responsible for advising on all aspects of the project.

## Competing interests

The authors declare no competing interests.
