## [Peer Review File · Communications Medicine]

Reviewers' comments:

Reviewer #1 (Remarks to the Author):

In this manuscript the authors describe the concern related to the use of wearables for scratch detection, since they lack the fundamental capacity of estimating also the scratch intensity. The authors propose the use of a multimodal ring device and machine learning algorithms to detect scratching behavior and, at the same time, quantify scratch intensity.

1. The ABSTRACT is well written and comprehensive. The problem is well formulated as the objectives of this study and the used methodology.

2. The INTRODUCTION is well written, the main problem is well described and supported by appropriate references. Also the objective is well described. In particular, it is well described the current state of the art and what is lacking in the currently available wearables devices.

Minor comment: Page 2, the last 4 lines and page 3 the last 10 lines of the paragraph should be added at the beginning of the method section since it is a description of the ring design.

3. The RESULTS are very well presented and also the images are well designed and easy to understand.

4. In the DISCUSSION the author introduced some interesting aspects regarding the future potential use of their wearable device to evaluate treatment efficacy and the possibility to automatically upload data to patient records in the cloud. These data, together with data relative to sleep, body temperature, medications use, etc... could in the future contribute to the development of personal treatments. They also well presented the limitations of the current wearables in use and how their product can contribute to overcome such limitations.

Reviewer #2 (Remarks to the Author):

General comments

The authors evaluate the performance of our algorithms on 20 individuals using Leave One Subject Out Cross Validation and using data from 14 additional participants, they show that their algorithms achieve clinically-relevant discrimination of scratching intensity levels. This work demonstrates that a finger-worn device can provide multi-dimensional, objective, real-time measures for the action of scratching. The main research findings of this paper will be important for the full understanding of measure the scratch intensity.

Specific comments

1, Can this device distinguish behaviors such as picking, kneading, and rubbing from scratching behaviors?

2, This device measures strength by specializing in scratching intensity. However, it is important to measure the scratch intensity when scratching the skin that actually has itching. Do you actually induce itch stimuli by such as histamine, cowhage or chloroquine to the skin and measure the intensity of scratching with this device?

This is an engineering driven manuscript on a multimodal sensing ring for quantification of scratch intensity. My critiques are below:

Major:

- A significant thrust of this paper’s novelty is the claim to measure scratch intensity. In fact, a prior paper has already reported on scratch intensity and correlated those measurements based on atopic dermatitis severity prior to the preprint of this article: *J Am Acad Dermatol.* 2023 Mar;88(3):726-729. doi: 10.1016/j.jaad.2022.09.032. Epub 2022 Sep 23. Thus, the authors should be more clear that this is not the first sensor to do so and explain their differentiation. See excerpt from the manuscript published online in 2022.

Fig 1. ** P -value $< .001$. * P -value $< .01$. Scratch events per sleep hour was computed for AD patients categorized by Validated Investigator Global Assessment (vIGA) status of 2, 3, or 4 across all sleep nights. Across all IGA levels, there was a statistically significant increase in scratch events per sleep hour. Average scratch intensity per night was computed for AD patients and categorized by vIGA status of 2, 3, or 4 across all sleep nights. With increasing vIGA scores, scratch intensity per night also increased. However, only scratch intensity per night for patients with vIGA 4 vs patients with vIGA 2 were noted to be statistically significant. Total sleep time in hours was computed for AD patients categorized by vIGA status of 2, 3, or 4 across all sleep nights. Total sleep time decreased with increased disease severity, although this was only statistically significant between vIGA 2 and 3, and vIGA 2 and 4. Wake after sleep onset (WASO) in minutes was computed for AD subjects categorized by vIGA status of 2, 3, or 4 across all sleep nights. The total amount of time spent awake after sleep onset increased with disease severity, with statistical significance between vIGA 2 and 3 and vIGA 2 and 4. vIGA, Validated Investigator Global Assessment; WASO, wake after sleep onset.

- It would be interesting to see how the placement of the ring affects scratch intensity measurements – does placement more proximal on the finger yield vastly different results? In Chun et al. *Science Advances* 2021 – there is an explanation on how signal changes with placement changes. This is important to increase confidence in the repeatability of the sensor’s outputs.
- Scratching behavior varies significantly between atopic dermatitis patients and healthy normals. Thus, the authors should not that this is a major limitation and future validation should be conducted in populations that suffer from real itch.
- I would be clear to readers in the Discussion that while the sensing element is mounted on the ring – there appears to be a bulky wrist mounted system that affects this current technology’s user acceptability. I do agree that finger mounted systems if able to

accommodate a battery to sufficiently power a microphone and accelerometer would be a nice form factor – however this was not achieved here.

Minor:

- Eczema is not the preferred clinical term – atopic dermatitis would be more accurate and specific.
- I do not agree that scratch intensity is closely tied to itch intensity -> if the author's can provide a citation; it maybe but whether itch intensity is more associated with scratch duration or scratch frequency versus scratch intensity has not been studied. While this seems to be nit picky, the terminology and description of scratch is relevant as outcome measurements for clinical trials.
- Scratch intensity and frequency of scratch are both causes – saying one above the other is over inflating scratch intensity as a variable.

Reviewer #4 (Remarks to the Author):

This paper summarizes a study with normal controls that evaluates a prototype of a wearable device (in the form of a finger ring) in detecting and assessing the intensity of scratching. In several health conditions, like eczema and psoriasis, tracking of itching and evaluating the intensity of scratching that may result in skin damage can help physicians evaluate treatment effectiveness.

Overall, the study is well thought out, the methodology valid, and the results obtained can potentially be useful to the research community considering the study attempts to quantify the intensity of scratching. However, there several issues that must be addressed before the paper is accepted for publication. One of the contributing factors is probably the use of the English language, which has limitations that make the paper difficult to follow when imprecise and often incorrect expressions and terminology are used. The following major and/or minor issues are presented simply in order of appearance in the text.

Specific issues:

— p2, last sentence: "...these sensors capture *low* frequency finger and arm motions..." but in the rest of the text frequencies in the range 150-400 Hz are used as features. How is it possible for a human to move their finger 400 times per second? In fact, on p.3 the analog reading of the microphone shows oscillations of a few cycles per second.

—p3. It is unclear if data from the first 20 subjects were used to develop (train) the model with data from the tablet and the next 14 subjects provide data to evaluate the model? Please clarify.

—p4. How are the 6,600 data samples computed? Recordings were split into 1-sec segments; each subject provided 30 minutes of recordings and there were 20 participants, i.e., $30 \text{ min} * 60 \text{ sec/min} * 20 \text{ subjects} = 36,000 \text{ data sample}$. Please clarify.

— p5. Next to last sentence: "transform each ...purposes."

- clarify what 'sample' is.

- is Savgol filter the Savitzky-Golay filter (Savgol is probably a Python function)?

- intuitively, it would make sense for vibrations to decrease, not increase, as force and friction increase; can you explain?

- How exactly were the subjects instructed to move their finger? For example, low, medium, high force first, and then in separate recordings low, medium, high speed? If that is the case, these events are independent and there are no nine combinations of force and speed as Fig. 4 implies. From a subject's point of view, it would be extremely difficult to do 9 combinations very accurately. If indeed subjects were doing 9 combinations of force and speed, could individual subject reproduce their own results? The tablet data could provide that information.

— p6. Fig.4. In this reviewer's opinion, it is a good practice to separate figures (which typically show graphs, plots, histograms, etc.) from tables (always show only formatted text) and not mix them up in one entity.

- Fig.4C. The signals shown are said to be normalized (zero mean, unit variance) before taking the FFT; how come there is still a huge DC component in the spectrum, for both the microphone and accelerometer data? If the time signals were zero mean, the DC component should be zero.

- The figure caption talks about an "ablation study" which was never discussed or explained in the text.

- It is unclear how it is possible to have so high frequencies (up to 400 Hz) with a human moving their finger.

— p7. The proposed method (combination of microphone and accelerometer data) uses the magnitude of the signal spectrum as features, but it is unclear what the "naïve predictor" uses as features.

- Also, how do you explain that the proposed method performs worse than the naïve predictor in the 61 – 180 mW range? According to Fig.4C, this would correspond to the first 6 combinations of force and speed (i.e., all low and medium force combinations).

—p13 The conversion of power to a scale between 1 and 10 is very useful, as is the inclusion of linear and nonlinear approaches. Can the authors provide more information on what the data showed, i.e., if one approach is more plausible than the other? The higher errors in prediction at higher powers might suggest that the process is indeed nonlinear.

—p13 can the proposed algorithms detect scratching from other movements in individual subjects?

— p15 Delete 3rd paragraph (repetition of the 2nd one).

— p16 How were the parameters of the Savitzky-Golay filter selected? Why 5th order and why 0.21 sec?

— p15-17, calculation of the velocity: first the removal of certain extrema that are thought of as outliers, although understandable, seems that forces smooth sinusoidal curves that go from peak to trough in every cycle and eliminates all the smaller fluctuations in between which may alter the actual cycle and velocity values. Then, in the next step, a straight line between successive extrema is drawn which replaces the sinusoidal curve with a sawtooth curve. This forces the instantaneous velocity to become uniform in the time interval between two extrema. But the sinusoidal nature of the recorded signal shows that velocity is actually slower around the extrema and much faster in the time points in between.

—p17. The first 400 values of the frequency magnitude are selected as features fed to the neural network, but one would expect most of these values to be zero considering they represent frequency of finger movement.

—p17. The neural network on p.17 is described as having 2 hidden layers of 1000 nodes each, whereas on p.18 as having 3 hidden layers of 1200 nodes each.

Overall impression: Potentially the proposed method might prove useful, but it still represents early work with a few issues:

— the study included only normal controls in a well-controlled environment, and the reference baseline was established by simulating scratching of a tablet, not real scratching of skin.

— the device used may be potentially useful, but it would require development and validation. In fact, the present study did not address the presence of noise/distortion in the data from the ad hoc hardware.

— the neural networks were trained with inputs of 575 features, which were simply the first 575 frequency values of signals obtained from the sensors; no feature selection or justification why these might be good features was provided.

— differentiation of scratching vs detection of other movements was not well described, especially if the methodology is to be used with patients during night sleep.

— Finally, a more thorough literature review would be helpful; e.g, <https://doi.org/10.1038/s41746-023-00821-y> and the citations it provides give a good review of past and present research.

Language issues:

The paper has limitations in language and should be reviewed by a native speaker to correct proper use of English.

Reviewers' Comments and Authors' Response

Paper Title: A Multimodal Sensing Ring for Quantification of Scratch Intensity

Authors: Akhil Padmanabha, Sonal Choudhary, Carmel Majidi, and Zackory Erickson

Reviewer #1

In this manuscript the authors describe the concern related to the use of wearables for scratch detection, since they lack the fundamental capacity of estimating also the scratch intensity. The authors propose the use of a multimodal ring device and machine learning algorithms to detect scratching behavior and, at the same time, quantify scratch intensity.

1. *The ABSTRACT is well written and comprehensive. The problem is well formulated as the objectives of this study and the used methodology.*

Authors: Thank you for your kind words.

2. *The INTRODUCTION is well written, the main problem is well described and supported by appropriate references. Also the objective is well described. In particular, it is well described the current state of the art and what is lacking in the currently available wearables devices.*

Minor comment: Page 2, the last 4 lines and page 3 the last 10 lines of the paragraph should be added at the beginning of the method section since it is a description of the ring design.

Authors: Thank you for your feedback. In terms of your minor comment, we are unsure if you are suggesting to move the 10 sentences to the methods or if you are suggesting that the methods section is missing details on the device design. In the case that it is the first of the two, we believe the brief description of the device is helpful for context before the reader reads the results and discussion sections. In the case that it is the second of the two, we have "Section 4.1 Device Design" which explains the design of the ring in detail.

3. *The RESULTS are very well presented and also the images are well designed and easy to understand.*

Authors: Thank you for your kind words.

4. *In the DISCUSSION the author introduced some interesting aspects regarding the future potential use of their wearable device to evaluate treatment efficacy and the possibility to automatically upload data to patient records in the cloud. These data, together with data relative to sleep, body temperature, medications use, etc... could in the future contribute to the development of personal treatments. They also well presented the limitations of the current wearables in use and how their product can contribute to overcome such limitations.*

Authors: Thank you for your kind words.

Reviewer #2

General comments

The authors evaluate the performance of our algorithms on 20 individuals using Leave One Subject Out Cross Validation and using data from 14 additional participants, they show that their algorithms achieve clinically-relevant discrimination of scratching intensity levels. This work demonstrates that a finger-worn device can provide multi-dimensional, objective, real-time measures for the action of scratching. The main research findings of this paper will be important for the full understanding of measure the scratch intensity.

Specific comments

1. *Can this device distinguish behaviors such as picking, kneading, and rubbing from scratching behaviors?*

Authors: Thank you for the question. It is difficult for us to say for these specific tasks/activities without further human testing, but we believe this could work based on our observations from other related activities. In order to definitively say, one could capture additional data of picking, kneading, and rubbing behaviors and retrain the classification model.

We added the following sentence to the Discussion section: “While our study focused on differentiating scratching behaviors from many common daily activities, future work could additionally consider behaviors that may be prevalent or similar to scratching, such as kneading, picking, and rubbing.”

2. *This device measures strength by specializing in scratching intensity. However, it is important to measure the scratch intensity when scratching the skin that actually has itching. Do you actually induce itch stimuli by such as histamine, cowhage or chloroquine to the skin and measure the intensity of scratching with this device?*

Authors: For the scope of this paper, we do not induce itch using stimuli and do not evaluate on patients with natural itch. The focus of this paper is primarily on the methods and algorithms of estimation of scratch intensity and we believe that future work is needed to explore experiments with inducing itch.

Reviewer #3 has a similar comment (#3) and we have incorporated the following in our discussion section:

“One limitation of this work is the lack of evaluation with patients with pruritus. Future work will focus on development of a fully integrated, wireless ring for continuous scratch monitoring allowing long term real world studies that evaluate our algorithms in

practice on representative patients, who may scratch differently than healthy participants.”

Reviewer #3

This is an engineering driven manuscript on a multimodal sensing ring for quantification of scratch intensity. My critiques are below:

Major:

1. A significant thrust of this paper's novelty is the claim to measure scratch intensity. In fact, a prior paper has already reported on scratch intensity and correlated those measurements based on atopic dermatitis severity prior to the preprint of this article: *J Am Acad Dermatol.* 2023 Mar;88(3):726-729. doi: 10.1016/j.jaad.2022.09.032. Epub 2022 Sep 23. Thus, the authors should be more clear that this is not the first sensor to do so and explain their differentiation. See excerpt from the manuscript published online in 2022.

Authors: Thank you for this feedback. We have now cited this paper and added the following text in the paper:

“A wearable device, placed on the back of the hand, has previously been introduced to estimate scratching force during sleep [25]. While this prior work lacks specific explanations regarding its methodologies, our system stands out due to its implementation of a multimodal sensing strategy with accelerometer and contact microphone data and its utilization of a pressure-sensitive tablet for training and validation, enabling us to attain statistically significant differentiation among various intensity levels.”

2. *It would be interesting to see how the placement of the ring affects scratch intensity measurements – does placement more proximal on the finger yield vastly different results? In Chun et al. Science Advances 2021 – there is a explanation on how signal changes with placement changes. This is important to increase confidence in the repeatability of the sensor's outputs.*

Authors: As seen in the screenshot below from Fig 1A, the prototype ring is longer in the proximal axis than a standard ring and it covers most of the proximal phalanx of the index finger. There are only a few millimeters it can be shifted up or down the proximal phalanx for most participants and during the human studies, we didn't control for the placement of the device on the proximal phalanx. Thus, we don't expect there to be a huge difference in signal quality anywhere on the proximal phalanx.

The ring is a tethered prototype and the primary technical contributions of the paper are the scratch intensity framework and methods, rather than the device. As a result, we didn't include any information on the change in signal strength as the location is modified.

We address this with the following change in the Discussion:

“Additional testing and evaluation of our multimodal wearable ring for tracking scratching behaviors would be helpful before long-term deployment, including a signal strength and noise analysis for when the ring is placed at different locations along the index finger or moved to other fingers.”

3. *Scratching behavior varies significantly between atopic dermatitis patients and healthy normals. Thus, the authors should not that this is a major limitation and future validation should be conducted in populations that suffer from real itch.*

Authors: Thanks, we agree. We have added the following in the Discussion section “One limitation of this work is the lack of evaluation with patients with pruritus. Future work will focus on development of a fully integrated, wireless ring for continuous scratch monitoring allowing long term real world studies that evaluate our algorithms in practice on representative patients, who may scratch differently than healthy participants.”

4. *I would be clear to readers in the Discussion that while the sensing element is mounted on the ring – there appears to be a bulky wrist mounted system that affects this current technology’s user acceptability. I do agree that finger mounted systems if able to accommodate a battery to sufficiently power a microphone and accelerometer would be a nice form factor – however this was not achieved here.*

Authors: We modified a sentence in our discussion to: “While our prototype device is currently tethered and requires a battery and supporting electronics on the wrist in addition to the sensorized ring, there are existing, well-adopted commercial ring devices, such as the Oura Ring which show that electronics with multiple sensors, including accelerometers, can be successfully embedded in a ring form factor with a long lasting battery life and aesthetic design.”

Minor:

1. *Eczema is not the preferred clinical term – atopic dermatitis would be more accurate and specific.*

Authors: Thanks, we have replaced all occurrences of “eczema” with “atopic dermatitis” in the manuscript.

2. *I do not agree that scratch intensity is closely tied to itch intensity -> if the author’s can provide a citation; it maybe but whether itch intensity is more associated with scratch duration or scratch frequency versus scratch intensity has not been studied. While this seems to be nit picky, the terminology and description of scratch is relevant as outcome measurements for clinical trials.*

Authors: Thanks, this is a fair point. We modified the following sentences to soften our claims:

“Scratch intensity ~~is closely tied~~ can be related to itch intensity, which is tracked by doctors and patients using scales and questionnaires such as the Peak Pruritus Numerical Rating Scale (PPNRS) and the 5-D Itch Scale.”

3. *Scratch intensity and frequency of scratch are both causes – saying one above the other is over inflating scratch intensity as a variable.*

Authors: We agree. We have softened our statements with the following:

“Scratch intensity is one of the primary factors that determines damage to skin and holds additional insight into the degree of disruption that itching has on everyday life and sleep.”

“In this work, we introduce a framework for quantification of scratch intensity, a the root cause of damage to skin due to chronic itch, and detection of scratching.”

Reviewer #4

This paper summarizes a study with normal controls that evaluates a prototype of a wearable device (in the form of a finger ring) in detecting and assessing the intensity of scratching. In several health conditions, like eczema and psoriasis, tracking of itching and evaluating the intensity of scratching that may result in skin damage can help physicians evaluate treatment effectiveness.

Overall, the study is well thought out, the methodology valid, and the results obtained can potentially be useful to the research community considering the study attempts to quantify the intensity of scratching. However, there several issues that must be addressed before the paper is accepted for publication. One of the contributing factors is probably the use of the English language, which has limitations that make the paper difficult to follow when imprecise and often incorrect expressions and terminology are used. The following major and/or minor issues are presented simply in order of appearance in the text.

Specific issues:

- 1. p2, last sentence: "...these sensors capture *low* frequency finger and arm motions..." but in the rest of the text frequencies in the range 150-400 Hz are used as features. How is it possible for a human to move their finger 400 times per second? In fact, on p.3 the analog reading of the microphone shows oscillations of a few cycles per second.*

Authors: Thank you for your comment, as it provides us an opportunity to improve clarity of the paper. To clarify, we are using features in the range of 0-399 Hz, not 150-400 Hz. In the methods section, we state the following:
"From c^G , we select the first 400 amplitudes, yielding contact microphone features, $c^G \in \mathbb{R}^{400}$, which corresponds to frequencies from 0 to 399 Hz. From a^G , we select the first 175 amplitudes, yielding accelerometer features, $a^G \in \mathbb{R}^{175}$, which corresponds to frequencies from 0 to 174 Hz. These features are concatenated, forming a feature vector, $G \in \mathbb{R}^{575}$."

We also note that while the accelerometer can capture finger motions, the contact microphone captures vibrations that permeate through the finger, and these contact vibrations (not finger motion) can certainly exceed frequencies in the 300+ Hz range. To improve clarity of this in the paper, in the "Scratch Intensity Dataset and Feature Extraction" section, we have added: "For the contact microphone data, we specifically use the first 400 amplitudes, which corresponds to frequencies from 0 to 399 Hz, while for the accelerometer data, we use the first 175 amplitudes yielding accelerometer features, which corresponds to frequencies from 0 to 174 Hz."

The analog reading shown on p3 is zoomed out which is why the higher frequency data is not discernible. We have modified Figure 1B to make this clearer as shown below.

2. *p3. It is unclear if data from the first 20 subjects were used to develop (train) the model with data from the tablet and the next 14 subjects provide data to evaluate the model? Please clarify.*

Authors: In the “Scratch Intensity Validation” section, we now state the following: “We trained a scratch intensity model using all 20 participants' data from the first human study with optimal model hyperparameters found using a grid search. We then validated the performance of this trained model through a second human study (Carnegie Mellon University IRB Approval under 2021_00000480) in which 14 healthy participants were instructed to perform various intensities of scratching across their body.”

3. *p4. How are the 6,600 data samples computed? Recordings were split into 1-sec segments; each subject provided 30 minutes of recordings and there were 20 participants, i.e., 30 min* 60 sec/min*20 subjects = 36,000 data sample. Please clarify.*

Authors: Thank you for the feedback. For clarity, the 30 minutes of data is the total amount of data for all 20 participants combined, not per participant. We have modified the paper to make this more clear.

To clarify, we modified the following sentence:

“We capture a total of 30 minutes of scratching behavior across all 20 participants (approximately 1.5 minutes of scratching per participant), in total, there is initially 30 minutes of data captured with all 20 participants, resulting in 6,600 data samples as seen in Fig. 4B.”

In Figure 4B, there is a further breakdown of the dataset as seen below:

B

Window Size	1 second
Stride	0.25 seconds
Number of Participants	20
Data per participant before cleaning	1.5 min, 333 samples
Total data before cleaning	6660 samples
Data per participant after cleaning	211.4 samples \pm 59.8 samples
Total data after cleaning	4227 samples
Mean Label	119.6 mW \pm 107.0 mW
Median Label	83.44 mW

4. *p5. Next to last sentence: “transform each ...purposes.” clarify what ‘sample’ is.*

Authors: As we mentioned the term ‘sample’ previously in the manuscript, we added the following sentence where it was first introduced.

“We split the data into 1 second windows of sensor measurements with an associated power label automatically generated via the Sensel Morph pressure tablet. Henceforth, we will refer to each 1 second window of data as a sample.”

5. *is Savgol filter the Savitzky-Golay filter (Savgol is probably a Python function)?*

Authors: Thank you for catching this. We have changed from “Savgol” to “Savitzky-Golay” throughout the manuscript.

6. *intuitively, it would make sense for vibrations to decrease, not increase, as force and friction increase; can you explain?*

Authors: Thank you for the question. With a constant speed, the amplitude of the vibrations increase as force and velocity increases. This is shown by Fig. 4C where the lines are clearly ordered by vibration amplitude based on the force applied by the participants.

In the paper, we have changed the following sentence,

"As force and velocity increase, the ~~signal amplitudes correspondingly increase as friction and~~ of the vibrations between the fingernail and skin ~~correspondingly~~ rise."

7. *How exactly were the subjects instructed to move their finger? For example, low, medium, high force first, and then in separate recordings low, medium, high speed? If that is the case, these events are independent and there are no nine combinations of force and speed as Fig. 4 implies. From a subject's point of view, it would be extremely difficult to do 9 combinations very accurately. If indeed subjects were doing 9 combinations of force and speed, could individual subject reproduce their own results? The tablet data could provide that information.*

Authors: In the "Section 2.1 Scratch Intensity Dataset and Feature Extraction", we state "They were led through the 9 combinations of low, medium, and high force, and low, medium, and high speed scratching on the Sensel Morph for 10 seconds each." The ordering was low force low speed, low force medium speed, low force high speed, medium force low speed, medium force medium speed, medium force high speed, high force low speed, high force medium speed, and high force high speed. This specific ordering helped participants distinguish all 9 combinations, and findings from our study, such as those seen in Fig. 4C, show that participants were indeed able to scratch at these different forces and velocities.

The reasoning for doing 9 combinations of varying force and speed was to obtain a diverse dataset with varying scratching behavior and with a wide range of power labels. To clarify this, we have added the following sentences in "Section 2.1 Scratch Intensity Dataset and Feature Extraction":

"The ordering of the combinations was low force low speed, low force medium speed, low force high speed, medium force low speed, medium force medium speed, medium force high speed, high force low speed, high force medium speed, and high force high speed. Participants have differing interpretations of low, medium, and high force and speed which allows us to collect a diverse dataset of varying scratching behavior with a large range of power labels."

8. *p6. Fig.4. In this reviewer's opinion, it is a good practice to separate figures (which typically show graphs, plots, histograms, etc.) from tables (always show only formatted text) and not mix them up in one entity.*

Authors: Thank you for this point. We certainly agree with this practice in most cases, but for this paper, we found it helpful to group our plots/tables so that each collection tells a complete story. We grouped by the following topics: scratch intensity results from the first human study (Figure 4), scratch intensity results from the second human study (Figure 5), scratch detection results (Figure 6).

9. *Fig.4C. The signals shown are said to be normalized (zero mean, unit variance) before taking the FFT; how come there is still a huge DC component in the spectrum, for both the microphone and accelerometer data? If the time signals were zero mean, the DC component should be zero.*

Authors: Normalization in this case refers to Min-Max Scaling to [0, 1]. Note that the plot is also smoothed using a Savitzky-Golay filter for visualization.

To clarify this, in "Section 2.1 Scratch Intensity Dataset and Feature Extraction" and in the caption of Fig. 4C, we have added the following:

"For each of these combinations, we transform each sample into the frequency domain, min-max normalize to scale values to between 0 and 1, average across all healthy participant data..."

"FFT features for contact microphone and accelerometer for combinations of force and velocity are min-max normalized to scale values to between 0 and 1, averaged across all healthy participant data, and subsequently smoothed using a Savitzky-Golay filter for visualization."

10. *The figure caption talks about an "ablation study" which was never discussed or explained in the text.*

Thank you for pointing this out. We have modified the following sentences:

"As shown in Fig.4G, we conduct an ablation study ~~Ablation results are additionally located in Fig. 4G~~ to quantify the benefits of multimodal sensing. For this ablation study, we evaluate our algorithm with just the contact microphone features and just the accelerometer features to compare the performance of single-sensor approaches with the multimodal model."

“Fig.6D also provides ablation results from an ablation study to quantify the benefits of the multimodal sensing ring”

11. *It is unclear how it is possible to have so high frequencies (up to 400 Hz) with a human moving their finger.*

Authors: Thanks for the question. We addressed this in response to your first comment.

12. *p7. The proposed method (combination of microphone and accelerometer data) uses the magnitude of the signal spectrum as features, but it is unclear what the “naïve predictor” uses as features.*

Authors: In “Section 2.2 Scratch Intensity Model Performance”, we state “All models perform better than the naive predictor which uses the mean, 119.64 mW, of all the labels as the prediction.”

We added the following sentence to the caption of Figure 4H for additional clarity, “**The naive predictor uses the mean of all labels as the prediction.**”

13. *Also, how do you explain that the proposed method performs worse than the naïve predictor in the 61 – 180 mW range? According to Fig.4C, this would correspond to the first 6 combinations of force and speed (i.e., all low and medium force combinations).*

Authors:

The naive predictor is designed specifically to work well on a small window of power ranges (61-180 mW) as it is the mean of all power labels in our dataset. In this range, it performs comparable to our approach. However, for all other ranges, it performs noticeably worse.

The 9 combinations can also span very large ranges of the power spectrum; as shown in Fig 5F, people have very different interpretations of intensity so the first 6 combinations of force and velocity span more than just 61-180 mW.

14. *p13 The conversion of power to a scale between 1 and 10 is very useful, as is the inclusion of linear and nonlinear approaches. Can the authors provide more information on what the data showed, i.e., if one approach is more plausible than the other? The higher errors in prediction at higher powers might suggest that the process is indeed nonlinear.*

Authors: For the data we collected, the linear model is superior to the nonlinear model presented as the mean absolute error is lower. However, we agree that at higher recorded powers, a nonlinear model could fit the data better.

In “Section 2.3 Scratch Intensity Validation”, we have added the following:

“In Supplementary Information Fig. S2, we additionally present an alternate non-linear ~~fitted square root function to convert the 0-600 mW power scale to a 0-10 continuous scale~~ which achieves a mean absolute error of 1.37 units.”

In “Section 3 Discussion”, we have added the following:

“Using the 0-10 objective, continuous scale and a linear scaling, our method yields a MAE of 0.96 units and with a non-linear scaling, our method yields a MAE of 1.37 units. For our data, the linear model is superior to the non-linear model presented but for higher recorded scratching power, a non-linear model could fit the data better.”

15. *p13 can the proposed algorithms detect scratching from other movements in individual subjects?*

Authors: Yes, as detailed in “Section 2.4 Scratch Detection Dataset and Feature Extraction” and in “Section 2.5 Scratch Detection Model Performance”, we collect a dataset with 20 participants which includes scratching on 7 different locations and 7 non-scratching activities and present results using leave one subject out cross validation (LOSO-CV). We utilize LOSO-CV as it mirrors the clinically relevant scenario of tracking scratching in new subjects whose data is not in the training set. We find that the average classification accuracy between the scratching and non-scratching behaviors using the multimodal model is 89.98%.

16. *p15 Delete 3rd paragraph (repetition of the 2nd one).*

Authors: Corrected. Thank you for catching this. Apologies for the error.

17. *p16 How were the parameters of the Savitzky-Golay filter selected? Why 5th order and why 0.21 sec?*

Authors: The parameters were chosen through manual fine-tuning of the parameters. Similar to the screenshot below from Figure 3, we plotted the Y position over time and the peaks and valleys, and randomly selected a subset of samples from each participant. We subjectively chose 5th order and 0.21 seconds after trying various

parameters and visually observing the peak and valleys to make sure they aligned with the movement of the centroid of the contact.

To clarify this in the manuscript, we added the following sentence:

“These parameters were chosen through visual observation of the sensed contacts and generated peaks and valleys from a randomly selected subset of samples from each participant.”

18. p15-17, calculation of the velocity: first the removal of certain extrema that are thought of as outliers, although understandable, seems that forces smooth sinusoidal curves that go from peak to trough in every cycle and eliminates all the smaller fluctuations in between which may alter the actual cycle and velocity values. Then, in the next step, a straight line between successive extrema is drawn which replaces the sinusoidal curve with a sawtooth curve. This forces the instantaneous velocity to become uniform in the time interval between two extrema. But the sinusoidal nature of the recorded signal shows that velocity is actually slower around the extrema and much faster in the time points in between.

Authors: This is a fair point. Our velocity algorithm is an estimation given the limitations of our pressure sensitive tablet. We previously attempted a method that calculated instantaneous velocity for each set of consecutive points in a sample and then averaged the instantaneous velocities across the entire sample. Nevertheless, this approach proved ineffective due to error in the Sensel Morph’s calculation of contact centroids, causing them to unpredictably shift and greatly impacting velocity calculations.

19. p17. The first 400 values of the frequency magnitude are selected as features fed to the neural network, but one would expect most of these values to be zero considering they represent frequency of finger movement.

Authors: Thanks for the comment. We have addressed this in response to your first comment.

20. *p.17. The neural network on p.17 is described as having 2 hidden layers of 1000 nodes each, whereas on p.18 as having 3 hidden layers of 1200 nodes each.*

Authors: Thanks for the observation. There are two neural networks presented, one for scratch intensity regression and one for scratch detection, as shown in Figure 2. On pg. 17, under the section “4.2 Scratch Intensity” and subsection “4.2.2 Dataset, Feature Extraction, and Model Design”, the neural network with 2 hidden layers of 1000 nodes each is for scratch intensity regression. On pg. 18, under the section “4.3 Scratch Detection Experimental Design” and subsection “4.3.1 Dataset, Feature Extraction, and Model Design”, the neural network with 3 hidden layers of 1200 nodes each is for scratch detection.

Overall impression: Potentially the proposed method might prove useful, but it still represents early work with a few issues:

21. *the study included only normal controls in a well-controlled environment, and the reference baseline was established by simulating scratching of a tablet, not real scratching of skin.*

Authors: Thanks for the feedback; these are valid points. We have included these as limitations in our discussion section as shown below:

“We showed that our scratch intensity model, trained on data collected from scratching on a tablet, accurately captures scratching on the skin. Although we have a pressure sensitive tablet for controlled validations, as a community, we lack the skin-tight pressure sensor array technology that would be necessary to quantify the error (in mW power units) that our method exhibits for on skin scratching.”

“One limitation of this work is the lack of evaluation with patients with pruritus. Future work will focus on development of a fully integrated, wireless ring for continuous scratch monitoring allowing long term real world studies that evaluate our algorithms in practice on representative patients, who may scratch differently than healthy participants.”

22. *the device used may be potentially useful, but it would require development and validation. In fact, the present study did not address the presence of noise/distortion in the data from the ad hoc hardware.*

Authors: The ring is a tethered prototype and the primary technical contributions of the paper are the scratch intensity framework and methods, not the device. As a result, we didn't include any information on the change in signal strength and noise as the location is modified.

We address this with the following change in the Discussion:

“Additional testing and evaluation of our multimodal wearable ring for tracking scratching behaviors would be helpful before long-term deployment, including a signal strength and noise analysis for when the ring is placed at different locations along the index finger or moved to other fingers.”

23. *the neural networks were trained with inputs of 575 features, which were simply the first 575 frequency values of signals obtained from the sensors; no feature selection or justification why these might be good features was provided.*

Authors: Thanks for the comment. We have provided justification of why frequency features are useful for scratch intensity regression in “Section 2.1 Scratch Intensity Dataset and Feature Extraction” in the paragraph starting with “We observe that power information is captured in the frequency domain of the contact microphone and accelerometer z-axis...”. While manual feature extraction may improve results slightly, we have found strong performance by providing all available features to our data-driven models.

24. *differentiation of scratching vs detection of other movements was not well described, especially if the methodology is to be used with patients during night sleep.*

Authors: Thank you, this is a very fair observation. The primary contribution of our work is scratch intensity regression, not scratch detection. With that said, we have found that a multimodal sensing approach can also aid with scratch detection.

25. *Finally, a more thorough literature review would be helpful; e.g., <https://doi.org/10.1038/s41746-023-00821-y> and the citations it provides give a good review of past and present research.*

Authors: Thank you. The linked paper was published a couple months after our submission. We have added it to our paper and additionally added the following citations:

- Lavery, Michael Joseph, et al. "Nocturnal pruritus: prevalence, characteristics, and impact on ItchyQoL in a chronic itch population." *Acta dermato-venereologica* 97.4 (2017): 513-515.
- Feuerstein, Johanna, et al. "Wrist actigraphy for scratch detection in the presence of confounding activities." *2011 Annual International Conference of the IEEE Engineering in Medicine and Biology Society*. IEEE, 2011.

26. Language issues: The paper has limitations in language and should be reviewed by a native speaker to correct proper use of English.

Authors: 3 of the 4 authors including the primary writer are native English speakers and were born and raised in the United States. The paper was also reviewed and edited by 3 other native English speakers before submission. We agree that certain parts of the paper can be made more clear but we have found no grammatical errors or other English language errors.

REVIEWERS' COMMENTS:

Reviewer #1 (Remarks to the Author):

My comments very addressed in a correct way, and I do not have further comments. I recommend the new version of the article for publication.

Reviewer #2 (Remarks to the Author):

The authors have responded appropriately to the points pointed out by the reviewers.

Reviewer #4 (Remarks to the Author):

In this revised version of the manuscript the authors have made a sincere effort to address all the issues raised previously.

The paper is now ready for publication.